# A Metal-Free, Disulfide Oxidized Form of Superoxide Dismutase 1 as a Primary Misfolded Species with Prion-Like Properties in the Extracellular Environments Surrounding Motor Neuron-Like Cells

**DOI:** 10.3390/ijms22084155

**Published:** 2021-04-16

**Authors:** Chika Takashima, Yasuhiro Kosuge, Masahisa Inoue, Shin-Ichi Ono, Eiichi Tokuda

**Affiliations:** 1Laboratory of Clinical Medicine, School of Pharmacy, Nihon University, 7-7-1 Narashinodai, Funabashi, Chiba 274-8555, Japan; phti15146@g.nihon-u.ac.jp (C.T.); ono.shinichi@nihon-u.ac.jp (S.-I.O.); 2Laboratory of Pharmacology, School of Pharmacy, Nihon University, 7-7-1 Narashinodai, Funabashi, Chiba 274-8555, Japan; kosuge.yasuhiro@nihon-u.ac.jp; 3Laboratory of Structure and Functional Research, Faculty of Pharmaceutical Science, Tokushima Bunri University, 180 Nishihamaboji, Yamashiro, Tokushima 770-8514, Japan; inoue@ph.bunri-u.ac.jp; 4Division of Neurology, Akiru Municipal Medical Center, 78-1 Hikida, Akiruno, Tokyo 197-0834, Japan

**Keywords:** amyotrophic lateral sclerosis, superoxide dismutase 1, extracellular environment, prion-like property, protein misfolding

## Abstract

Superoxide dismutase 1 (SOD1) is a metalloenzyme with high structural stability, but a lack of Cu and Zn ions decreases its stability and enhances the likelihood of misfolding, which is a pathological hallmark of amyotrophic lateral sclerosis (ALS). A growing body of evidence has demonstrated that misfolded SOD1 has prion-like properties such as transmissibility between cells and intracellular propagation of misfolding of natively folded SOD1. Recently, we found that SOD1 is misfolded in the cerebrospinal fluid of sporadic ALS patients, providing a route by which misfolded SOD1 spreads via the extracellular environment of the central nervous system. Unlike intracellular misfolded SOD1, it is unknown which extracellular misfolded species is most relevant to prion-like properties. Here, we determined a conformational feature of extracellular misfolded SOD1 that is linked to prion-like properties. Using culture media from motor neuron-like cells, NSC-34, extracellular misfolded wild-type, and four ALS-causing SOD1 mutants were characterized as a metal-free, disulfide oxidized form of SOD1 (apo-SOD1^S-S^). Extracellular misfolded apo-SOD1^S-S^ exhibited cell-to-cell transmission from the culture medium to recipient cells as well as intracellular propagation of SOD1 misfolding in recipient cells. Furthermore, culture medium containing misfolded apo-SOD1^S-S^ exerted cytotoxicity to motor neuron-like cells, which was blocked by removal of misfolded apo-SOD1^S-S^ from the medium. We conclude that misfolded apo-SOD1^S-S^ is a primary extracellular species that is linked to prion-like properties.

## 1. Introduction

Amyotrophic lateral sclerosis (ALS) is an incurable neurodegenerative disease characterized by progressive loss of upper and lower motor neurons in the central nervous system (CNS) [1]. The loss of the motor neurons compromises voluntary movements of skeletal and respiratory muscles [1]. The majority (~90%) of ALS cases are sporadic with unknown etiology, whereas the remining cases are familial [1]. To date, over 30 causative genes have been identified in familial ALS (ALSoD, https://alsod.ac.uk/, accessed on 16 April 2021). Similar to other neurodegenerative diseases, proteins encoded by the ALS-causing genes undergo aberrant changes in their structure and are intracellularly deposited in misfolded and aggregated states, comprising the major pathological hallmark of ALS [2].

The clinical symptoms of most ALS patients start focally and spread contiguously to other regions over the course of disease progression [3]. Such pathological and clinical features of ALS resemble those of prion diseases, a group of transmissible and lethal neurodegenerative diseases caused by misfolded prion proteins [4]. A growing body of evidence supports that misfolded/aggregated superoxide dismutase 1 (SOD1), the first identified gene of familial ALS [5], has prion-like properties. Cell culture experiments have revealed that intracellular misfolded and aggregated SOD1 is actively released or leaked from cells and transferred to neighboring cells [6,7]. Then, the internalized SOD1 acts as a conformational template or a propagated seed to convert natively folded SOD1 into a misfolded/aggregated state [6,7,8,9]. The prion-like properties of misfolded/aggregated SOD1 are applicable to *in vivo* conditions. The transmissibility of misfolded/aggregated SOD1 and acceleration of the ALS-like phenotype can be replicated by inoculation of protein extracts from the spinal cords of mouse models [10,11,12,13] or patients with ALS carrying *SOD1* mutations [14]. Taken together, misfolded/aggregated SOD1 spreads its cytotoxicity to cellular and histological regions affected by ALS.

Because extracellular spaces are present between cells, misfolded/aggregated SOD1 can exist in extracellular fluids of the CNS such as cerebrospinal fluid (CSF). Indeed, we have recently reported that SOD1 was misfolded, but not aggregated or oligomeric, in CSF from sporadic ALS patients as well as familial ALS patients carrying *SOD1* mutations [15,16]. Moreover, exposure of CSF from sporadic ALS patients induces cell death in motor neuron-like NSC-34 cells, and its toxicity is caused by misfolded SOD1 [16]. However, it is not known how the presence of extracellular misfolded SOD1 in the CSF reflects the pathogenesis of ALS within motor neurons. In general, proteins and metabolic wastes from the CNS parenchyma are transported to the interstitial fluid, and this extracellular fluid covers cells in the CNS [17]. Interstitial fluid drains along the capillary and perivascular spaces of arteries, and then reaches the CSF [17]. Thus, the following two possibilities exist for the origin of extracellular misfolded SOD1 in CSF: (i) Misfolded SOD1 is directly secreted by motor neurons into the extracellular environment or (ii) SOD1 is released in a conformation other than the misfolded state, and subsequently, SOD1 becomes misfolded during circulation in the CNS. Regarding the former possibility, both wild-type and mutant SOD1 have been reported to be secreted by primary cultured neurons and neuron-like cell lines into culture medium [7,18], designated here as conditioned medium. Moreover, extracellular mutant SOD1 in donor-conditioned medium induces prion-like propagation of misfolding of natively folded SOD1 in recipient cells [7]. Thus, misfolded SOD1 species with prion-like properties are most likely secreted by motor neurons. Nevertheless, little is known about the conformational features of the extracellular misfolded SOD1 species that play a central role in prion-like transmission and propagation.

The aim of the present study was to characterize the conformational features of extracellular misfolded SOD1 in conditioned medium from a motor neuron model of ALS and to determine which extracellular misfolded species has prion-like properties, including cell-to-cell transmission and intracellular seeding activity.

## 2. Results

### 2.1. Generation of Motor Neuron Models of ALS with Intracellular Accumulation of Misfolded and Aggregated SOD1

To generate motor neuron models of ALS in which misfolded and aggregated SOD1 accumulates inside cells, expression vectors containing human wild-type SOD1 (hSOD1^WT^) or four ALS-causing hSOD1 mutants (G93A, G37R, D90A, and G85R) tagged with green fluorescent protein (GFP) were used to transiently transfect NSC-34 cells, a motor neuron-like cell line. Among the ALS-linked mutants used here, G93A, G37R, and D90A are wild-type-like mutants that retain enzymatic activity [19,20], but their conformational stability is decreased [21]. G85R is a metal-binding site mutant that compromises the affinity for Cu and Zn ions, resulting in no enzymatic activity [22]. After 72 h of transfection, Western blot analysis with anti-SOD1 antibody showed that hSOD1-GFP-fused proteins were expressed in the non-ionic detergent Nonidet-P-40 (NP-40) soluble fraction from the cells transfected with GFP-tagged hSOD1^WT^ or hSOD1 mutants (Figure 1a). We observed a faster migration for G85R and D90A than hSOD1^WT^, which is consistent with previous studies [22,23]. Fluorescence microscopy revealed that the ALS-causing hSOD1 mutants were aggregated in the cytoplasm of NSC-34 cells (Figure 1b). This finding was confirmed by Western blots on the NP-40 insoluble faction, which revealed the hSOD1 species presented as insoluble aggregates in mutated hSOD1-GFP cells (Figure 1a). Furthermore, intracellular misfolded SOD1 was detected in the NP-40 soluble fraction from hSOD1^WT^ and the ALS-linked hSOD1 mutant cells using a sandwich enzyme-linked immunosorbent assay (ELISA) with antibodies specific for misfolded SOD1 (Figure 1c). EDI recognizes misfolded SOD1 with a solvent-exposed dimer interface [24]. C4F6 reacts with misfolded SOD1 with a destabilized small hairpin loop [25,26]. Collectively, we successfully generated motor neuron models of ALS in which misfolded and aggregated SOD1 was intracellularly deposited.

### 2.2. Total Protein Level of Extracellular Human and Murine SOD1 in Conditioned Medium

Natively folded SOD1 is characterized by binding of Cu and Zn ions, and also by formation of an intramolecular disulfide bond between Cys57 and Cys146 [27]. Thus, the occurrence of SOD1 misfolding could be associated with dysregulation of post-translational modifications. We investigated the Cu/Zn binding state and thiol/disulfide redox balance in extracellular misfolded SOD1. Before we characterized the conformational features of misfolded SOD1, first, we quantified the total protein level of extracellular SOD1. The NSC-34 cells were transiently transfected with GFP-tagged hSOD1^WT^ or the ALS-linked hSOD1 mutants. After 72 h of transfection, the conditioned medium was collected and analyzed to quantify the total protein level of hSOD1 using Western blot assays (Figure 2a). While the level of extracellular hSOD1^WT^ was 3.0 ng/µg of medium protein, that of all tested mutants was around 0.3 ng/µg of medium protein (G93A, 0.25 ng; G37R, 0.23 ng; D90A, 0.32 ng; and G85R, 0.31 ng) (Figure 2b). This decrease in the extracellular mutant hSOD1 level is consistent with the results of a previous study [28]. Interestingly, no change was observed in the level of endogenous murine SOD1 in the medium regardless of the ALS-linked hSOD1 expression (Figure 2a). The total extracellular murine SOD1 protein level in all tested media was approximately 0.24 ng/µg of medium protein (Figure 2c). These results suggest that the secretory pathway of hSOD1 differs from that of endogenous murine SOD1 and that the hSOD1 pathway is inhibited when the ALS-causing hSOD1 mutants are intracellularly expressed. We excluded the possibility that there was little contamination of cell lysis or debris in the conditioned medium because β-tubulin, an intracellular protein that is not secreted to extracellular space, was not detected in the medium (Figure 2a).

### 2.3. Involvement of Extracellular Oxidative Stress in the Thiol/Disulfide Redox Balance of Extracellular hSOD1

While the formation of an intramolecular disulfide bond in SOD1 confers conformational stability to the protein, reduction of the disulfide bond in SOD1 promotes the likelihood of misfolding [29,30]. Thus, we monitored the thiol/disulfide redox balance in extracellular hSOD1. The conditioned medium was treated with 100 mM iodoacetamide to prevent cysteine resides of SOD1 from undergoing artificial oxidation, and the results were analyzed by non-reducing Western blots (Figure 3a). In extracellular hSOD1^WT^, the ratios of the disulfide (S-S) and thiol (-SH) forms to the total hSOD1^WT^ level were 72% and 28%, respectively (Figure 3b). In the extracellular hSOD1 mutants, the disulfide oxidized form (S-S) was common to all tested mutants with an abundance of 99% relative to the total hSOD1 level in each mutant (Figure 3b). We confirmed that the disulfide oxidized form was a primary extracellular species of mutant SOD1 because the disulfide reduced form was not observed even in the concentrated conditioned medium (data not shown).

On the basis of the increased fraction of the disulfide oxidized form of hSOD1 in the conditioned medium, we hypothesize that the extracellular environment around the NSC-34 cells expressing hSOD1 mutants shifts toward oxidative stress conditions. Therefore, we measured the levels of 4-hydroxynonenal (4-HNE) and lipid peroxides produced under oxidative stress conditions [31] in the conditioned media (Figure 3c). Western blot analysis showed that the level of 4-HNE-conjugated proteins was approximately 2.0-fold greater in medium of all tested SOD1 mutants than in the no vector or GFP medium (Figure 3d). Interestingly, the conditioned medium from the hSOD1^WT^ cells also exhibited a 1.5-fold higher level of 4-HNE-conjugated proteins than that from the no vector or GFP cells (Figure 3d). These results suggest that the extracellular environment surrounding the NSC-34 cells expressing hSOD1 shifts towards oxidative stress conditions, and such conditions are involved in the increase in the fraction of the disulfide oxidized form of hSOD1.

### 2.4. The Fraction of Non-Natively Folded Human SOD1, but not Murine SOD1, Is Increased in the Extracellular Environment

Natively folded SOD1, but not other SOD1 conformations, has enzymatic activity [32,33], indicating that both Cu/Zn binding and disulfide bond formation are essential to exert SOD1 activity. Thus, the level of active SOD1 corresponds to that of natively folded SOD1. Considering the presence of other SOD isozymes in the conditioned medium, we carefully quantified the extracellular level of active SOD1 using a SOD Assay Kit-WST. The extracellular level of active hSOD1^WT^ was 1.9 ng/µg of medium protein (Figure 4a), which was 64% of the total extracellular hSOD1^WT^ level. By contrast, in the wild-type-like hSOD1 mutants, the extracellular level of active hSOD1 was only around 0.1 ng/μg of medium protein (G93A, 0.11 ng; G37R, 0.10 ng; and D90A, 0.17 ng) (Figure 4a). Despite the involvement of disulfide bonds in SOD1 stability, the abundance of active hSOD1 mutants was only around 50% (G93A, 44%; G37R, 43%; and D90A, 53%), suggesting that the remaining hSOD1 mutants, which contain a disulfide bond (G93A, 56%; G37R, 57%; and D90A, 47%), are enzymatically inactive. A metal-deficient mutant, G85R, completely lacked enzymatic activity even though its disulfide bond was oxidized (Figure 4a). Thus, the extracellular environment surrounding cells expressing the ALS-linked hSOD1 contains an increased fraction of non-natively folded or inactive hSOD1 mutants.

We validated the increased fraction of inactive hSOD1 by treating the conditioned medium with 1 mM CuSO_4_, 1 mM ZnSO_4_, or both. Treatment with both Cu and Zn ions restored the extracellular levels of active hSOD1^WT^ and the wild-type-like hSOD1 mutants, but not of G85R which lost the ability to Cu/Zn coordination (Figure 4c). However, treatment with either Cu or Zn ions alone did not affect the active hSOD1 level (Figure 4c), suggesting that extracellular inactive or non-natively folded hSOD1 is characterized by a Cu and Zn-deficient state.

Notably, the level of active murine SOD1 was approximately 0.24 ng/µg of medium protein in all tested media regardless of whether hSOD1 was expressed (Figure 4b), and the proportion of active murine SOD1 relative to total murine SOD1 was nearly 100%, indicating that Cu and Zn ions were fully charged in extracellular murine SOD1.

### 2.5. Metal-Free, Disulfide Oxidized SOD1 (apo-SOD1^S-S^) Is the Primary Misfolded Species in Conditioned Medium

We quantified the misfolded SOD1 level in conditioned medium using an ELISA with EDI or C4F6 antibodies. The level of EDI-positive species in the hSOD1^WT^ medium was 0.079 ng/µg of medium protein, whereas that in the ALS-causing hSOD1 medium was over 0.11 ng/µg of medium protein (G93A, 0.123 ng; G37R, 0.129 ng; D90A, 0.110 ng; and G85R, 0.181 ng) (Figure 5a). The same results were observed for the quantification of C4F6-positive species (hSOD1^WT^, 0.079 ng; G93A, 0.136 ng; G37R, 0.130 ng; D90A, 0.114 ng; and G85R, 0.191 ng) (Figure 5b). No misfolded SOD1 was found in the medium from no vector and GFP cells (Figure 5a,b). These results are consistent with our observation, showing that all extracellular murine SOD1 is natively folded (Figure 4b). Using the quantitative data on the active hSOD1 level (Figure 4a), we calculated the abundance of natively folded and misfolded hSOD1 in the medium. For extracellular hSOD1^WT^, the ratios of natively folded and misfolded hSOD1^WT^ to total hSOD1^WT^ were 64% and 3%, respectively (Figure 5d). Although the remaining 33% of hSOD1^WT^ species (referred to here as to other conformations) were enzymatically inactive, they were not misfolded, suggesting that they might be precursors of misfolded species. The extracellular ALS-causing hSOD1 consisted of a smaller proportion of natively folded protein, and also a larger proportion of misfolded protein as compared with hSOD1^WT^ (Figure 5d).

To clarify whether extracellular misfolded SOD1 could be metal-free, the conditioned medium was treated with both 1 mM CuSO_4_ and 1 mM ZnSO_4_ for 24 h. Then, the extracellular level of misfolded SOD1 was measured using an ELISA. As expected, treatment of the medium with Cu and Zn ions abolished the ELISA signals for misfolded SOD1 (Figure 5a,b, gray), indicating that extracellular misfolded SOD1 lacks Cu and Zn ions. The total extracellular hSOD1 level was not changed by Cu and Zn treatment of the medium (Figure 5c, gray). Moreover, we determined the thiol/disulfide redox state of extracellular misfolded SOD1 using immunoprecipitation of the conditioned medium with C4F6 followed by non-reducing Western blotting (Figure 5e). Extracellular misfolded hSOD1^WT^ and all tested hSOD1 mutants were both detected as the disulfide oxidized form (S-S). Collectively, extracellular misfolded hSOD1 species are characterized as the metal-free, disulfide oxidized form of hSOD1 (apo-SOD1^S-S^).

We explored the possibility of the presence of additional extracellular hSOD1 conformations such as oligomers and aggregates. However, the conditioned medium did not contain oligomeric SOD1 species that were intermolecularly crosslinked with cysteine residues of the protein when the medium was treated with iodoacetamide and analyzed by non-reducing Western blotting (data not shown). Furthermore, insoluble SOD1 aggregates were not observed because the abundance pattern of insoluble SOD1 was not changed when the medium was centrifuged at high speed or filtered with a 0.22 µm membrane filter (Appendix A). These results suggest that oligomeric and aggregated SOD1 species were not present in the conditioned medium from our motor neuron model of ALS.

### 2.6. Extracellular Misfolded apo-SOD1^S-S^ Induces Intracellular Propagation of hSOD1^WT^ Misfolding

We characterized the conformational features of extracellular misfolded hSOD1 and termed this apo-SOD1^S-S^ (Figure 5). To address whether misfolded apo-SOD1^S-S^ could be transmissible between cells and induce intracellular propagation of hSOD1 misfolding in recipient cells, our first approach was to use conditioned medium from the NSC-34 cells expressing GFP-tagged hSOD1^WT^ or the ALS-causing hSOD1 mutants. The medium was added to the recipient NSC-34 cells transfected with GFP-untagged hSOD1^WT^. After 24 h of exposure, immunoprecipitation of the NP-40 soluble fraction from the recipient cells with C4F6 revealed that medium containing misfolded hSOD1^WT^ or hSOD1 mutants was sufficient to induce misfolding of intracellular hSOD1^WT^ (Figure 6a,b). This intracellular propagation was partially blocked by pretreatment of the medium with C4F6 to remove extracellular apo-SOD1^S-S^ (Figure 6a,b). Additionally, propagation of intracellular hSOD1^WT^ misfolding by the medium was suppressed by pretreatment of the recipient cells with wortmannin or 5-(*N*-ethyl-*N*-isopropyl) amiloride (EIPA), inhibitors of liquid-phase endocytosis (also known as macropinocytosis) (Figure 6c,d), indicating that extracellular misfolded apo-SOD1^S-S^ has the ability to undergo cell-to-cell transmission.

Our second approach was to use purified hSOD1^WT^ proteins with a disulfide bond. hSOD1^WT^ proteins with distinct metal contents (Cu/Zn SOD1^S-S^, Zn SOD1^S-S^, and apo-SOD1^S-S^) were used to treat recipient NSC-34 cells expressing GFP-tagged hSOD1^WT^. After 24 h of exposure of the purified proteins, immunoprecipitation of the NP-40 soluble fraction from the recipient cells with C4F6 displayed that apo-SOD1^S-S^ was able to elicit misfolding of intracellular hSOD1^WT^-GFP (Figure 6e,f). Cu/Zn SOD1^S-S^ (natively folded) and Zn SOD1^S-S^ did not have the ability to undergo prion-like propagation (Figure 6e,f). These data indicate that extracellular apo-SOD1^S-S^ could be a primary misfolded species that is linked to the prion-like properties of misfolded SOD1.

### 2.7. Prion-Like Propagation of Extracellular Misfolded apo-SOD1^S-S^ Induces Cell Death in Motor Neuron-Like NSC-34 Cells

To determine whether intracellular propagation of hSOD1 misfolding by extracellular misfolded apo-SOD1^S-S^ could be cytotoxic, recipient NSC-34 cells expressing GFP-untagged hSOD1^WT^ were exposed to the conditioned medium for 24 h, and cell viability was assessed using a Cell Counting Kit-8 assay (Figure 7a). The viability of the recipient hSOD1^WT^ cells treated with normal cultured medium was approximately 90% of that of untransfected NSC-34 control cells (Figure 7a), suggesting that intracellular expression of hSOD1^WT^ alone drops in their viability. The medium from NSC-34 cells expressing hSOD1^WT^ or the ALS-causing hSOD1 mutants, but not GFP alone, decreased the viability of hSOD1^WT^ recipient cells (Figure 7a, white). This lower viability was associated with the presence of extracellular misfolded apo-SOD1^S-S^ in the medium because removal of apo-SOD1^S-S^ from the medium by immunoprecipitation with C4F6 rescued cell viability (Figure 7a, black). Attenuation of cell viability was not associated with treatment with mouse IgG because the medium treated with normal mouse IgG did not increase its cell survival (Figure 7a, gray). 

We also measured the amounts of released lactate dehydrogenase (LDH), as an indicator of cytotoxicity, in hSOD1^WT^ recipient cells treated with the medium containing extracellular misfolded apo-SOD1^S-S^ (Figure 7b). In agreement with the decreased viability of the cells (Figure 7a, white), the medium that was not treated with normal mouse IgG or C4F6 exerted toxicity to the hSOD1^WT^ cells (Figure 7b, whit*e*). This cytotoxicity was inhibited when misfolded apo-SOD1^S-S^ was removed by immunoprecipitation with C4F6 (Figure 7b, black). Treatment of the medium with normal mouse IgG did not suppress its cytotoxicity (Figure 7b, gray). These results suggest that the cytotoxicity of extracellular misfolded apo-SOD1^S-S^ is related to prion-like propagation of SOD1 misfolding.

## 3. Discussion

The concept of prion-like properties of misfolded SOD1 has the potential to explain why the clinical phenotype of ALS spreads from an initial site of onset to contiguous anatomical regions in the CNS [3]. A better understanding of the nature of the prion-like properties of misfolded SOD1 is crucial to inhibit the spread of the disease. Several lines of evidence have demonstrated that misfolded SOD1 species exist in the extracellular environment, including conditioned medium, and that extracellular misfolded SOD1 is internalized into recipient cells when they are exposed to the conditioned medium [6,7], facilitating intracellular propagation of misfolding of endogenous native SOD1 [6,7,8,9]. In this study, we extended our understanding of the conformational features of extracellular misfolded SOD1 species, which are linked to prion-like properties. We found that extracellular misfolded SOD1 exists as a metal-free and disulfide oxidized form (apo-SOD1^S-S^) in the conditioned medium from NSC-34 cells expressing hSOD1^WT^ or the ALS-causing hSOD1 mutants (G93A, G37R, D90A, and G85R) (Figure 5a,b,e). Interestingly, the conformational features of extracellular misfolded SOD1 that we identified here correspond to those of misfolded SOD1 in CSF from sporadic and familial ALS patients carrying *SOD1* mutations [16]. Thus, it is possible that the extracellular experimental system (i.e., conditioned medium) used here seems to replicate the CSF in ALS patients to some extent. If so, the findings indicate that extracellular misfolded SOD1 circulating in the CNS of ALS patients has prion-like properties.

The conformational stability and integrity of a protein are highly influenced by the environment surrounding the protein. We observed that the extracellular environment around cells expressing hSOD1 shifted toward an oxidative stress condition. The conditioned medium from hSOD1^WT^ or the ALS-linked mutants cells contained a significantly increased level of 4-HNE-conjugated proteins (Figure 3c,d). Such an oxidative stress condition supports the increased fraction of the disulfide oxidized form (S-S) of hSOD1 in the conditioned medium (Figure 3a,b and Figure 5e). Interestingly, CSF from sporadic ALS patients has a markedly increased level of lipid peroxides and oxidized proteins [34,35], indicating that the extracellular environment shifts toward oxidative stress conditions, even in ALS patients. Furthermore, a cysteine residue of SOD1 is oxidatively modified in CSF from sporadic ALS patients, indicating an increased fraction of sulfenylated (-SOH) [36] and sulfonylated (-SO_3_H) [16] SOD1 in the extracellular environment of ALS patients. Hence, a strong oxidative environment likely triggers the formation of disulfide bonds as an oxidative modification of SOD1. Regarding exposure of SOD1 to oxidative stress, we recently reported that H_2_O_2_-induced oxidative stress provokes demetallation and misfolding of natively folded hSOD1^WT^ [37]. This evidence supports our observation, demonstrating that a part of the hSOD1 population loses Cu/Zn ions under an oxidative extracellular environment (Figure 4c).

An important finding of this study was that misfolded apo-SOD1^S-S^ is a primary extracellular species that is linked to prion-like properties (Figure 6). However, a previous *in vitro* study demonstrated that one conformational feature of misfolded hSOD1, which acts as an intracellular propagation seed, is a disulfide reduced form (SOD1^SH^), regardless of the Zn binding content [38]. Moreover, conditioned medium containing the misfolded hSOD1^G127X^ mutant, which lacks disulfide bonds and metal-binding ability (apo-SOD1^SH^) [19,39], can induce intracellular propagation of misfolding of native hSOD1^WT^ in recipient HEK293 cells [7]. Thus, there is a discrepancy in the thiol/disulfide state in misfolded SOD1 with prion-like properties between our findings and previous studies. This discrepancy may be explained by the following proposal. The redox balance of extracellular wild-type SOD1 and wild-type-like mutants (i.e., G93A, G37R, and D90A) shifts toward a disulfide oxidized form (S-S) under oxidative stress conditions (Figure 3c,d), whereas hSOD1^G127X^ still presents as a disulfide reduced form (-SH) even under oxidative conditions because of the lack of the Cys146 residue involved in the formation of the disulfide bond. Extracellular misfolded apo-SOD1^S-S^ is internalized by either endocytosis (Figure 6c,d) [6,9,40,41] or exosomes [7] into recipient cells. The internalized misfolded apo-SOD1^S-S^ has been shown to escape from endocytic/exosome vesicles into the cytosol [6,9], where natively folded SOD1, which acts as a substrate for the seeding reaction, is abundantly expressed. The reducing condition in the cytosol allows the disulfide bond of apo-SOD1^S-S^ to be reduced [42], and subsequently, apo-SOD1^SH^ acts as a pathological seed to propagate misfolding of native SOD1 [38]. Collectively, extracellular misfolded apo-SOD1^S-S^ does not seem to exhibit prion-like propagation in oxidative extracellular environments, but rather cell entry of apo-SOD1^S-S^ is an essential step to convert the reduced form and induce the propagation of native SOD1 misfolding.

To date, more than 180 mutations in the *hSOD1* gene have been found in familial ALS patients (ALSoD, https://alsod.ac.uk/, accessed on 16 April 2021). These ALS-causing hSOD1 mutants are categorized into the following five groups on the basis of the mutational sites related to hSOD1 stability [21]: (i) wild-type-like mutants (e.g., G93A, G37R, and D90A), (ii) metal-binding site mutants (e.g., G85R), (iii) disulfide forming cysteines mutants (e.g., C146R), (iv) dimer interface mutants (e.g., A4V), and (v) C-terminally truncated mutants (e.g., G127X). These divergent properties of hSOD1 mutants likely raise concerns about whether the misfolded apo-SOD1^S-S^ identified here (Figure 4 and Figure 5) could be a common extracellular species of hSOD1 mutant other than four mutants used here (G93A, G37R, D90A, and G85R). One study supports our observation, reporting that wild-type-like hSOD1^C111Y^ exists as apo-SOD1^S-S^ in CSF from familial ALS patients [16]. In contrast, hSOD1^G127X^, a C-terminally truncated and disulfide reduced (-SH) mutant [19,39], presents as apo-SOD1^SH^ in conditioned medium [7]. Furthermore, unlike our finding, extracellular hSOD1^G127X^ assembles into large aggregates, which is reasonable because apo-SOD1^SH^, but not apo-SOD1^S-S^, is a precursor of aggregates [43,44]. Collectively, the biophysical properties of each hSOD1 mutant highly influence its conformational state, even in extracellular environments.

We discovered that the conditioned medium containing misfolded apo-SOD1^S-S^ killed NSC-34 cells expressing hSOD1^WT^ (Figure 7a,b, white). The cytotoxicity is derived from the prion-like properties of extracellular misfolded apo-SOD1^S-S^ because removal of the extracellular species from the medium rescued cell viability (Figure 7a, black) or inhibited cytotoxicity (Figure 7b, black). These results are consistent with our recent study, which showed that extracellular misfolded SOD1 in CSF from sporadic ALS patients exerts cytotoxicity [16]. On the basis of our findings, we speculate that apo-SOD1^S-S^ is a primary misfolded species that exerts cytotoxicity. This speculation is supported by a series of experiments with recombinant misfolded apo-SOD1^S-S^, demonstrating that it includes cell death in primary spinal cord [45] or neuron-like cells [37,46]. The molecular mechanisms underlying the cytotoxicity of extracellular misfolded apo-SOD1^S-S^ that have been proposed include that apo-SOD1^S-S^ sequesters Zn ions in culture medium and decreases Zn bioavailability, leading to dysregulation of Zn homeostasis [37,46,47]. Thus, in addition to prion-like intracellular propagated misfolding of SOD1, the cytotoxic site of action of misfolded SOD1 is present in the extracellular environment.

As a therapeutic strategy, an approach targeting extracellular misfolded apo-SOD1^S-S^ would be beneficial because this protein propagates misfolding of intracellular native SOD1 (Figure 6) and induces cell death in motor neuron-like cells (Figure 7). Considering that conversion of SOD1 folding to misfolding is a reversible process [48], we offer an approach to decrease extracellular misfolded SOD1 by insertion of Cu/Zn ions into apo-SOD1^S-S^ to facilitate the folding process. In this regard, it is noteworthy that treatment of the conditioned medium with Cu and Zn ions succeeded in decreasing the misfolded SOD1 level (Figure 5a,b). Another approach to modify misfolded apo-SOD1^S-S^ is to use Cu^II^ (atsm), a Cu coordinating compound that directly inserts Cu ions into SOD1 [49]. Treatment with Cu^II^ (atsm) increased SOD1 activity and decreased misfolded SOD1 in a cellular model [50] and in mouse models of ALS carrying *SOD1* mutants [51]. Furthermore, Cu^II^ (atsm) treatment significantly prolonged survival in mouse models of ALS-SOD1, even when the treatment was initiated during the symptomatic stage [52,53]. Despite the therapeutic benefits of Cu^II^ (atsm), it is still unclear which SOD1 conformations of Cu/Zn binding and thiol/disulfide states directly interact with Cu^II^ (atsm). Future studies will be required to determine whether misfolded apo-SOD1^S-S^ can directly interact with Cu^II^ (atsm).

## 4. Materials and Methods

### 4.1. Expression Vectors

Wild-type SOD1 was cloned into pCMV3-C-GFP Spark^®^ from human cDNA (HG11727-ACG, Sino Biological, Beijing, China), and the C-terminus of SOD1 was tagged with GFP. Introduction of a point mutation (G93A,G37R, D90A, or G85R) into the human *SOD1* gene was performed by site-directed mutagenesis, and the correct DNA sequence was confirmed (Genewiz, Saitama, Japan). For experiments on intracellular propagation of hSOD1^WT^ misfolding by the conditioned medium, wild-type SOD1 with untagged GFP was also cloned into pCMV3 from human cDNA, (HG11727-UT, Sino Biological, Beijing, China). The plasmid DNA was amplified in Competent Quick *E. coil* DH5α (TOYOBO, Osaka, Japan), purified by a GenElute™ Endotoxin-free Plasmid Miniprep Kit (Sigma-Aldrich, St. Louis, MO, USA), concentrated using ethanol precipitation, and resolved in endotoxin- and DNase-free water (Nacalai Tesque, Kyoto, Japan). The concentrations of plasmid DNA were determined using a NanoDrop spectrophotometer (Thermo Fisher Scientific, Waltham, MA, USA).

### 4.2. Cell Culture and Transfection

NSC-34 cells, a murine motor neuron-like cell line generated by fusion of neuroblastoma cells with motor neuron-enriched embryonic spinal cords [54], were a kind gift from Professor Neil Cashman (Department of Medicine, Centre for Brain Health, University of British Colombia, Vancouver, BC, Canada).The NSC-34 cells (7.5 × 10^4^ cells/mL) were maintained in 60 mm culture dishes (Thermo Fisher Scientific, Waltham, MA, USA) in Dulbecco’s modified Eagle’s medium (DMEM) supplemented with 10% (*v*/*v*) fetal bovine serum (Biowest, Nuaillé, France) and 4.5 g/L *D*-glucose (Nacalai Tesque, Kyoto, Japan), in the absence of any antibiotics, under a humidified atmosphere of 5% CO_2_ at 37 °C. The cells were routinely passaged every 3 days because the doubling time of this cell line is 33 h.

For induction of hSOD1 expression, the cells (3 × 10^5^ cells/mL) were transiently transfected with plasmid DNA (1 µg/µL) encoding GFP-tagged hSOD1 or GFP alone using Lipofectamine 2000 (1 µg/µL), according to the manufacturer’s instructions (Thermo Fisher Scientific), placed in 100 mm culture dishes (Thermo Fisher Scientific, Waltham, MA, USA) in Opti-MEM™ reduced serum medium containing GlutaMAX™ (Gibco, Waltham, MA, USA). After 5 h of transfection with plasmid DNA, the Opti-MEM™ reduced serum medium was replaced with a mixture of 1:1 DMEM and Ham’s F-12 medium plus GlutaMAX™ (Gibco, Waltham, MA, USA) supplemented with 1% (*v*/*v*) fetal bovine serum, 0.1 mM non-essential amino acids, and 5 µM all *trans*-retinoic acid (Fujifilm Wako Pure Chemical Corporation, Osaka, Japan). After 72 h of transfection, the conditioned medium was collected and centrifuged at 1,500× *g* for 5 min at 4 °C to remove the suspended cells. Part of the collected medium was immediately treated with 100 mM iodoacetamide at 4 °C for 1 h to prevent the cysteine residues of SOD1 from undergoing artificial oxidation [19,55]. The NSC-34 cells attached to the culture dishes were removed using cell scrapers (Greiner, Frickenhausen, Germany), washed three times with sterilized phosphate-buffered saline (PBS) (pH 7.2) (Nacalai Tesque, Kyoto, Japan), and centrifuged at 1,500× *g* for 5 min at 4 °C. The collected media and cells were stored at −80 °C until use.

### 4.3. Fluorescent Imaging

The NSC-34 cells were plated onto glass coverslips that were coated with poly-*D*-Lysine (Matsunami Glass Ind., Osaka, Japan) in 24-well plates (Falcon, Corning, NY, USA) prior to transfection with hSOD1-GFP or GFP alone, as described above. After 72 h of transfection, the cells were fixed with PBS (pH 7.4) containing 4% (*w*/*v*) paraformaldehyde for 10 min at room temperature. Following brief washes with PBS (pH 7.4), images of the cells expressing hSOD1-GFP were taken at 30× magnification on a BZ-700X all-in-one florescence microscope (Keyence, Osaka, Japan). 

### 4.4. Ultrafiltration of Conditioned Medium

The original conditioned medium contained a small amount of protein (approximately 450 µg/mL). To increase the amount of protein obtained, the medium was concentrated using ultrafiltration devices (Amicon Ultra, molecular mass cut off <10 kDa, Millipore, Burlington, MA, USA). The original medium was added to the filter device and centrifuged at 10,000× *g* for 10 min at 4 °C. After four rounds of centrifugation, the concentrated medium inside the filter device was collected. The protein concentration of the concentrated medium was quantified using the Protein Assay Rapid Kit Wako II (Fujifilm Wako Pure Chemical Corporation, Osaka, Japan). The yield of concentrated medium was approximately 8100 µg/mL of protein, which was 18-fold higher than that of the original medium.

### 4.5. Protein Extraction

To extract proteins from the NSC-34 cells, they were sonicated using a Handy Sonic (UR-21P, TOMY SEIKO, Tokyo, Japan) for 1 min (three cycles of a 20-sec sonication and a 10-sec cooling on ice) in ice-cold PBS (pH 7.0) containing 1% (*v*/*v*) NP-40, 100 mM iodoacetamide, 5 mM EDTA, 10% (*v*/*v*) glycerol, 1× EDTA-free Complete Protease Inhibitor Cocktail (Roche Life Science, Penzberg, Germany), and 1× PhosSTOP (Sigma-Aldrich, St. Louis, MO, USA). The protein extracts were placed on crushed ice for 15 min and centrifuged at 20,000× *g* for 20 min at 4 °C. The supernatants of the extracts were collected as the NP-40 soluble fraction. The remaining pellets were washed three times with two volumes of lysis buffer and subjected to centrifugation at 20,000× *g* for 10 min at 4 °C. The pellets were sonicated and resolved in PBS (pH 7.0) containing 2% (*w*/*v*) sodium dodecyl sulfate (SDS) and designated as the NP-40 insoluble fraction. The protein concentration of the NP-40 soluble fraction was quantified using the Protein Assay Rapid Kit Wako II (Fujifilm Wako Pure Chemical Corporation, Osaka, Japan). 

### 4.6. Western Blotting

The NP-40 soluble fraction (20 µg/lane), the NP-40 insoluble faction (equivalent of 20 µg/lane), and the conditioned medium (4 µg/lane) were denatured with Laemmli sample buffer containing 6.7% (*v*/*v*) beta-mercaptoethanol (β-ME) at 37 °C for 10 min and separated using 10% (*v*/*v*) or 12.5% (*v*/*v*) polyacrylamide gels at 150 V for 1.5 h. The proteins were transferred onto polyvinylidene difluoride membranes (pore size 0.45 µm, GE Healthcare, Chicago, IL, USA) at 350 mA at 4 °C for 1.5 h. The membranes were blocked with PBS (pH 7.4) containing 3% (*w*/*v*) dried milk and 0.05% (*v*/*v*) Tween 20 for 1 h at room temperature. The membranes were incubated with primary antibodies against SOD1 (0.05 µg/mL, sc-11407, Santa Cruz Biotechnology, Dallas, TX, USA), GFP (0.02 µg/mL, sc-9996, Santa Cruz Biotechnology, Dallas, TX, USA), or 4-HNE (0.5 µg/mL, MHN-100P, Japan Institute for the Control of Aging, Shizuoka, Japan) at 4 °C for 18 h. After four washes with PBS (pH 7.4) containing 0.05% (*v*/*v*) Tween 20, the membranes were treated with horseradish peroxidase (HRP)-conjugated anti-IgG antibodies corresponding to the host animal in which the primary antibody was raised (1:10,000. A9044 or A9169, Sigma-Aldrich, St. Louis, MO, USA). The membranes were reacted with ImmunoStar LD (Fujifilm Wako Pure Chemical Corporation, Osaka, Japan), and the antigen-antibody complex signals were visualized using an LAS-1000 plus analyzer (GE Healthcare, Chicago, IL, USA). β-tubulin (×10,000, T-4026, Sigma-Aldrich, St. Louis, MO, USA) and Ponceau S staining (Intégral Corporation, Tokushima, Japan) were used as loading controls for the NP-40 soluble fraction and conditioned medium, respectively.

In-gel reduction was performed to analyze the thiol/disulfide redox balance of hSOD1, as described elsewhere [19,55]. In short, the conditioned medium (4 µg/lane) was prepared with Laemmli sample buffer in the absence β-ME. The samples were subjected to SDS-polyacrylamide gel electrophoresis (PAGE) under non-reducing conditions. The gels were warmed in a microwave oven for 20 sec at 500 W and incubated with running buffer for SDS-PAGE containing 1% (*v*/*v*) β-ME for 15 min at room temperature. The following procedures were the same as those used for Western blotting.

### 4.7. Enzyme-Linked Immunosorbent Assay

Misfolded SOD1 was quantified using a sandwich ELISA, as described previously with slight modifications [56]. As capture antibodies, two sets of antibodies specific for misfolded SOD1, EDI (0.5 µg/mL, SPC-206D StressMarq Biosciences, Victoria, BC, Canada) [24] and C4F6 (×2000, MM-0070-2-P, Médimabs, Montreal, QC, Canada) [45], were coated onto 96-well plates (Nunc-Immuno Module plate, Nunc, Roskilde, Denmark) at 4 °C for 18 h. The wells of the plates were washed three times with PBS (pH 7.4) containing 0.05% (*v*/*v*) Tween 20 and 5 mM EDTA, and then blocked with PBS (pH 7.4) containing 3% (*w*/*v*) bovine serum albumin and 5 mM EDTA at room temperature for 1 h. The NP-40 soluble faction (40 µg protein/well) or conditioned medium (40 µg protein/well) was added to the wells and incubated at room temperature for 1 h. To determine the absolute level of misfolded SOD1, metal-free hSOD1 with disulfide bonds (apo-SOD1^S-S^) was used as a standard to create a calibration curve. Sheep polyclonal anti-SOD1 antibody (0.5 µg/mL, 574,597, Sigma-Aldrich, St. Louis, MO, USA) was used as a detection antibody and incubated at 4 °C for 18 h. The detection antibody was labeled with HRP-coupled anti-sheep IgG (×2000, 1,721,017, Bio-Rad, Hercules, CA, USA) at room temperature for 1 h. Then, 100 mM citrate buffer (pH 5.0) containing 1 mg/mL *O*-phenylenediamine dihydrochloride (Fujifilm Wako Pure Chemical Corporation) and 0.03% (*v*/*v*) H_2_O_2_ was used as a color reagent, and the color reaction was stopped by adding 2 M HCl. Absorbance was measured at a wavelength of 490 nm using a microplate reader (FLUOstar Omega, BMG LabTech, Ortenberg, Germany). 

For quantification of the total protein level of extracellular hSOD1, an antibody that specifically reacts with human SOD1, but not murine SOD1, was used as a capture antibody (0.2 µg/mL, sc-17767, Santa Cruz Biotechnology, Dallas, TX, USA). A calibration curve was prepared using Cu/Zn hSOD1^WT^ (Sigma-Aldrich, St. Louis, MO, USA) as a standard.

### 4.8. Measurement of SOD1 Enzymatic Activity

The extracellular SOD1 activity in the ultrafiltration-concentrated medium was measured using a SOD Assay Kit-WST (Dojindo Laboratories, Kumamoto, Japan), according to the manufacturer’s directions. The technique is based on the inhibition rate of 2-(4-iodophenyl)-3-(4-nitrophenyl)-5-(2,4-disulfophenyl)-2*H*-tetrazolium and monosodium salt reduction. Because this assay was not able to discriminate SOD1 (Cu/Zn SOD) from the other SOD isozymes including SOD2 (Mn SOD) and SOD3 (EC SOD), the other isozymes were removed from the conditioned medium by the following treatments. First, SOD3 was removed from the medium using a Dynabeads™ Protein G Immunoprecipitation Kit (Invitrogen, Carlsbad, CA, USA) with an anti-SOD3 antibody (0.4 µg/mL, sc-271170, Santa Cruz Biotechnology, Dallas, TX, USA). Then, the medium without SOD3 was treated with 10 mM potassium cyanide for 15 min at 4 °C to inhibit SOD1 activity and to directly measure SOD2 activity [57]. Total SOD1 activity (both hSOD1 and endogenous murine SOD1) was calculated by subtracting the SOD2 activity from the total activity measured. For measurement of murine SOD1 activity, hSOD1 was removed from the medium without SOD3 by immunoprecipitation with a specific antibody against hSOD1 (0.5 µg/mL, sc-17767, Santa Cruz Biotechnology, Dallas, TX, USA). For quantification of extracellular active SOD1, Cu/Zn hSOD1^WT^, isolated from human erythrocytes (Sigma-Aldrich, St. Louis, MO, USA), was used as a standard to create a calibration curve. One unit of SOD1 activity was defined as the amount of SOD1 that inhibited the reduction of WST-1 by 50%. 

To add Cu and Zn ions to extracellular hSOD1 in vitro, part of the ultrafiltration-concentrated medium was incubated with 1 mM CuSO_4_, 1 mM ZnSO_4_, or both metal ions at 4 °C for 24 h. The level of active SOD1 was measured by the above-mentioned SOD Assay Kit-WST.

### 4.9. Immunoprecipitation

Misfolded SOD1 was detected using a Dynabeads™ Protein G Immunoprecipitation Kit (Invitrogen, Carlsbad, CA, USA), according to the manufacturer’s protocol with slight modifications. C4F6 (1:100) or normal mouse IgG (0.2 µg/mL, sc-2025, Santa Cruz Biotechnology, Dallas, TX, USA) was coupled with Dynabeads™ Protein G at 4 °C for 24 h. The Dynabeads-coupled antibodies were placed in a magnetic stand and washed with the provided washing buffer. The Dynabeads-C4F6 complex was mixed with the conditioned medium (1 mL) or the NP-40 soluble fraction (50 µg of proteins) at 4 °C for 24 h. Then, the Dynabeads-C4F6 misfolded SOD1 complex was placed in the magnetic stand. The misfolded SOD1 species captured by the Dyanabeads-C4F6 were eluted, and the eluates were analyzed by Western blotting under reducing or non-reducing conditions.

### 4.10. Intracellular Propagation of hSOD1^WT^ Misfolding by Conditioned Medium and Purified hSOD1^S-S^ Proteins

Conditioned medium from 72 h post-transfected NSC-34 cells expressing GFP-tagged hSOD1^WT^, the ALS causing hSOD1 mutants, or GFP alone was prepared as a donor medium. The NSC-34 cells (3 × 10^5^ cells/mL) expressing GFP-untagged hSOD1^WT^ were used as the recipient cells and exposed to the conditioned media in 6-well plates (Falcon, Corning, NY, USA) for 24 h. Intracellular prion-like propagation of misfolding of GFP-untagged hSOD1^WT^ in the recipient cells was evaluated by immunoprecipitation with C4F6 (1:100). To clarify the effects of endocytosis on cell-to-cell transmission of extracellular misfolded SOD1, the NSC-34 cells expressing GFP-untagged hSOD1^WT^ were pretreated with 50 nM wortmannin (Millipore, Burlington, MA, USA) or 100 µM EIPA (Cyaman Chemical, Ann Arbor, MI, USA), which are inhibitors of macropinocytosis, for 1 h at 37 °C before addition of the conditioned medium. Intracellular misfolded GFP-untagged hSOD1^WT^ in recipient cells treated with the endocytosis inhibitors was detected by immunoprecipitation with C4F6 (1:100).

For experiments on prion-like propagation of purified hSOD1^WT^ species, hSOD1^WT^ with disulfide bonds and different Cu/Zn ion contents was used. Cu/Zn hSOD1^WT^, isolated from human erythrocytes, was purchased from Sigma-Aldrich. Demetallation of Cu/Zn hSOD1^WT^ was performed by a Micro Float-A-Lyzer (molecular weight cut off 8–10 kDa, Repligen, Waltham, MA, USA) with a dialysis buffer (pH 3.0) containing 50 mM sodium acetate, 100 mM sodium chloride, and 5 mM EDTA, for 24 h, at 4 °C. Subsequently, the apo-hSOD1^WT^ protein solution was replaced with physiological buffer (pH 7.4) containing 50 mM *N*-(2-hydroxyethyl) piperazine-*N’*-2-ethanesulfonic acid (HEPES) and 100 mM sodium chloride. Then, apo-hSOD1^WT^ was incubated with one molar equivalent concentration of ZnSO_4_ to prepare Zn-binding hSOD1^WT^ (30 μM apo-SOD1^S-S^ protein was incubated with 30 μM ZnSO_4_). The prepared hSOD1^WT^ proteins (Cu/Zn SOD1^S-S^, Zn SOD1^S-S^, and apo-SOD1^S-S^) were sterilized by Millex^®^ syringe filters (pore size 0.22 µm, Millipore, Burlington, MA, USA) to remove microorganisms, dust, and preformed aggregates. The recipient GFP-tagged hSOD1^WT^ cells were exposed to 0.5 µg/mL wild-type hSOD1^S-S^ with various Cu/Zn contents for 24 h. 

### 4.11. Assessment of Viability and Cytotoxicity of Cells Exposed to Conditioned Medium Containing Misfolded apo-SOD1^S-S^

The NSC-34 cells (3.0 × 10^4^ cells/well) were transiently transfected with hSOD1^WT^ in 96-well plates (Falcon, Corning, NY, USA) for 48 h and used as recipient cells. The conditioned medium, in which NSC-34 cells expressing GFP-untagged hSOD1^WT^ or the ALS-linked hSOD1 mutants were cultured for 72 h, was added to the recipient cells. After 24 h of incubation, cell viability and cytotoxicity were evaluated using a Cell Counting Kit-8 (Dojindo Laboratories, Kumamoto, Japan) and a LDH Assay Kit-WST (Dojindo Laboratories, Kumamoto, Japan), according to the manufacturer’s protocol, respectively. To clarify the effects of extracellular misfolded apo-SOD1^S-S^ on cell viability and cytotoxicity, the misfolded species was removed from the conditioned medium by immunoprecipitation with C4F6 (1:100). As a control, the medium was also immunoprecipitated with normal mouse IgG (0.2 µg/mL). The recipient cells expressing GFP-untagged hSOD1^WT^ were exposed to the conditioned medium without misfolded apo-SOD1^S-S^ for 24 h, and their cell viability and cytotoxicity were assessed using the Cell Counting Kit-8 assay and the LDH release assay, respectively.

### 4.12. Statistical Analysis

The results are indicated as the mean ± SD. All statistical tests were analyzed using a Statcel 4 software (OMS Publishing, Saitama, Japan). For comparisons between two groups, the normality of the data sets was first determined. For normally distributed data, the homoscedasticity was determined using an F-test. Homoscedastic data were analyzed using a two-tailed Student’s t-test. Multiple group comparisons were performed using one-way ANOVA followed by the Tukey–Kramer post-hoc test. The significance level was defined as *p* < 0.05. All experiments were repeated at least twice. The number of samples in each experiment is described in each figure legend.

## Figures and Tables

**Figure 1 ijms-22-04155-f001:**
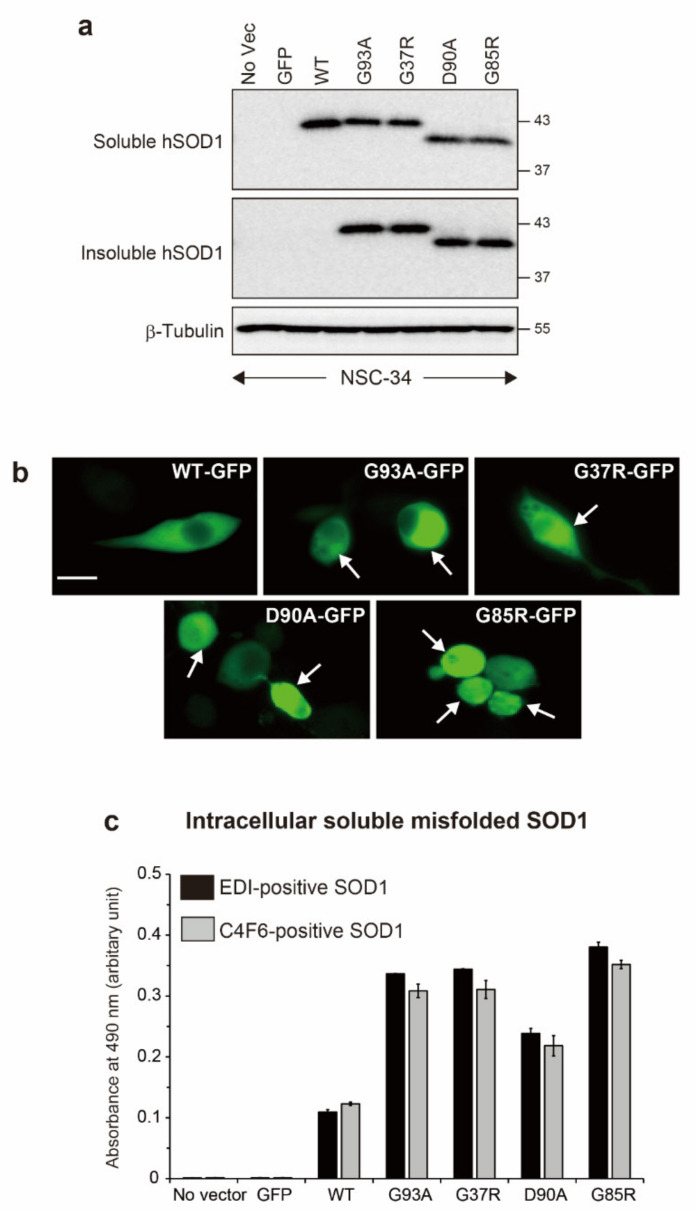
Generation of motor neuron models of ALS in which misfolded and aggregated hSOD1 are intracellularly deposited. NSC-34 cells were transiently transfected with expression vectors encoding human wild-type superoxide dismutase 1 (hSOD1^WT^) or the ALS-causing hSOD1 mutants (G93A, G37R, D90A, and G85R) tagged with green fluorescent protein (GFP), treated with all *trans*-retinoic acid, and cultured for 72 h. (**a**) Western blots of hSOD1 in the Nonidet P-40 (NP-40) soluble and insoluble factions from NSC-34 cells, β-tubulin was used as a loading control. No Vec., untransfected cells; (**b**) florescence microscopy of cells transfected with hSOD1^WT^ GFP or the ALS-linked hSOD1 mutant GFP. Arrows indicate protein aggregations containing hSOD1 GFP. Scale bar = 10 µm; (**c**) enzyme-linked immunosorbent assay signals of (black) EDI-positive and (gray) C4F6-positive misfolded SOD1 in the NP-40 soluble fraction. All data are shown as the mean ± SD, *n* = 3 in each transfection.

**Figure 2 ijms-22-04155-f002:**
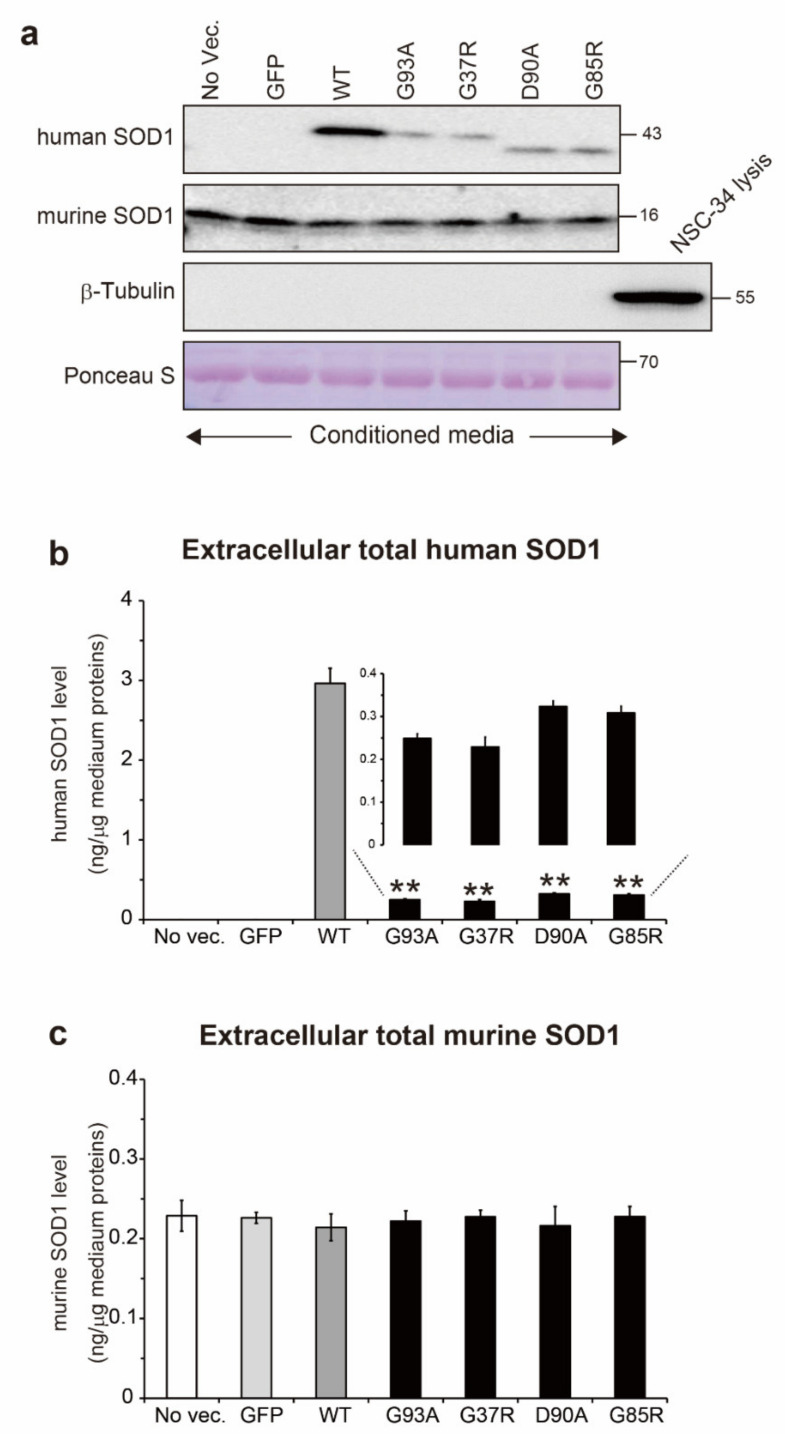
Decrease in the total level of extracellular hSOD1 mutants, but not murine SOD1, in the conditioned medium. The NSC-34 cells were transiently transfected with human wild-type superoxide dismutase 1 (hSOD1^WT^) and the ALS-causing hSOD1 mutants (G93A, G37R, D90A, and G85R) fused with green fluorescent protein (GFP). After 72 h of transfection, the conditioned medium from the cells was collected; (**a**) Western blots of hSOD1-GFP and endogenous murine SOD1 in the medium, β-tubulin was used as a marker for intracellular non-secreted protein to confirm whether cell lysis or debris could be contaminated in the conditioned medium. The amount of protein loaded for sodium dodecyl sulfate polyacrylamide gel electrophoresis was validated by Ponceau S staining; (**b**,**c**) quantification of the absolute total protein level of extracellular (**b**) hSOD1 and (**c**) murine SOD1. Data are shown as the mean ± SD, *n* = 3 in each transfection. Statistical analysis was performed using one-way ANOVA followed by the Tukey–Kramer post hoc test. ** *p* < 0.01 vs. hSOD1^WT^.

**Figure 3 ijms-22-04155-f003:**
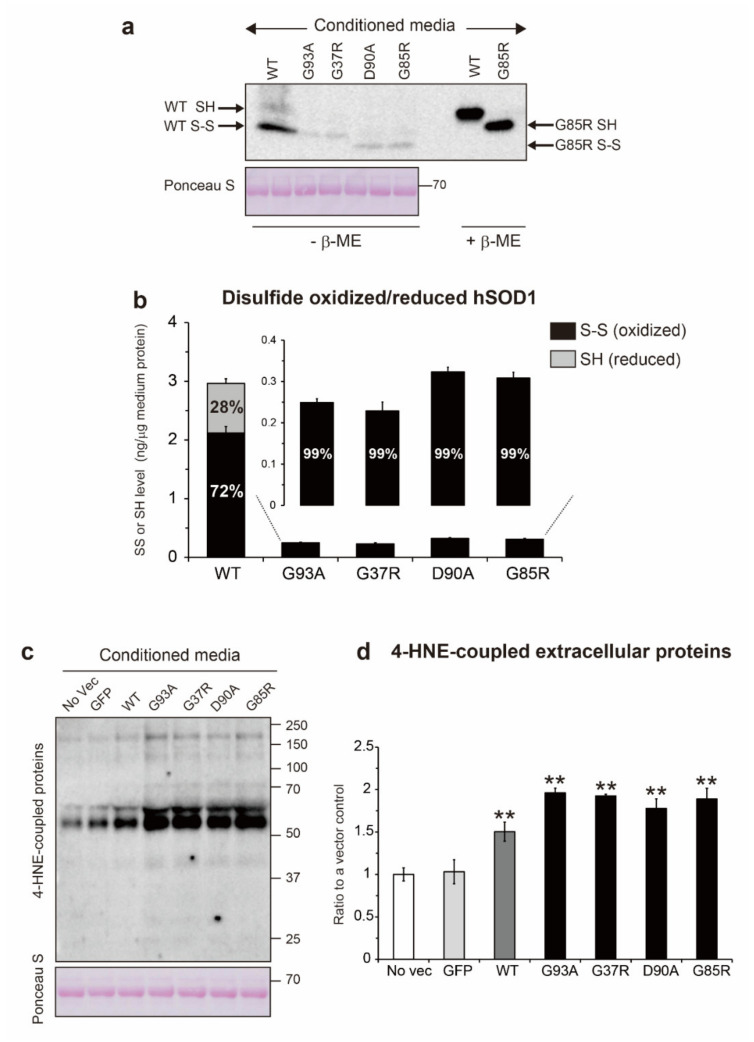
The thiol/disulfide redox balance of extracellular hSOD1 shifts toward oxidation because of increased oxidative stress in the conditioned medium. The conditioned medium of the NSC-34 cells expressing human wild-type superoxide dismutase 1 (hSOD1^WT^)-green fluorescence protein (GFP), the ALS-linked hSOD1-GFP, or GFP alone was treated with 100 mM iodoacetamide to block artificial oxidation of the thiol group of proteins. (**a**) Non-reducing Western blots of disulfide oxidized (S-S) and reduced (-SH) forms of hSOD1-GFP in the medium. β-ME = beta-mercaptoethanol; (**b**) quantification of absolute levels of (black) hSOD1^S-S^ and (gray) hSOD1^SH^ in the medium; (**c**) Western blots of 4-hydroxynonenal (4-HNE)-conjugated proteins in the conditioned medium; (**d**) densitometrical calculation of relative levels of 4-HNE-conjugated proteins in the medium. The amount of protein loaded for sodium dodecyl sulfate polyacrylamide gel electrophoresis was validated by Ponceau S staining. Data are expressed as the mean ± SD. Statistical analysis was performed using one-way ANOVA followed by the Tukey–Kramer post hoc test, *n* = 3 for each transfection. ** *p* < 0.01 vs. no vector.

**Figure 4 ijms-22-04155-f004:**
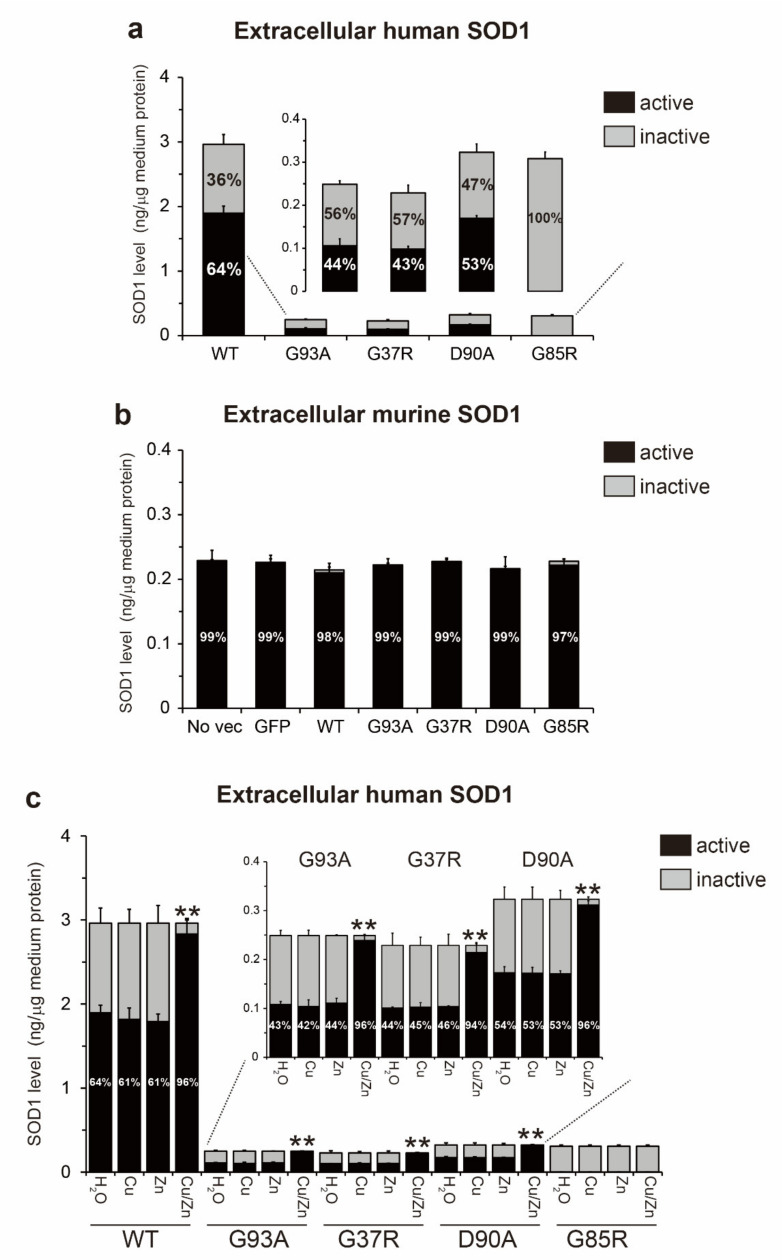
The population of extracellular Cu/Zn-deficient hSOD1 was increased in the conditioned medium. The conditioned medium from the NSC-34 cells transfected with human wild-type superoxide dismutase 1 (hSOD1^WT^)-green fluorescence protein (GFP), the ALS-linked hSOD1-GFP, or GFP alone, was treated with 100 mM iodoacetamide to block artificial oxidation of the thiol group of proteins. Then, extracellular human and murine SOD1 enzymatic activity in the medium was measured by a SOD Assay Kit-WST. The level of active SOD1 in the medium was calculated based on a calibration curve that was generated from a Cu/Zn SOD1 standard. The extracellular level of active (**a**) hSOD1 and (**b**) endogenous murine SOD1 in the medium; (**c**) the medium was treated with 1 mM CuSO_4_, 1 mM ZnSO_4_, or both, for 24 h, and active hSOD1 levels in the medium were quantified using a SOD Assay Kit-WST. All data are expressed as the mean ± SD. Statistical analysis was performed using one-way ANOVA followed by the Tukey–Kramer post hoc test, *n* = 3 for each treatment or for each transfection. ** *p* < 0.01 vs. H_2_O-added medium that corresponds to the same transfection.

**Figure 5 ijms-22-04155-f005:**
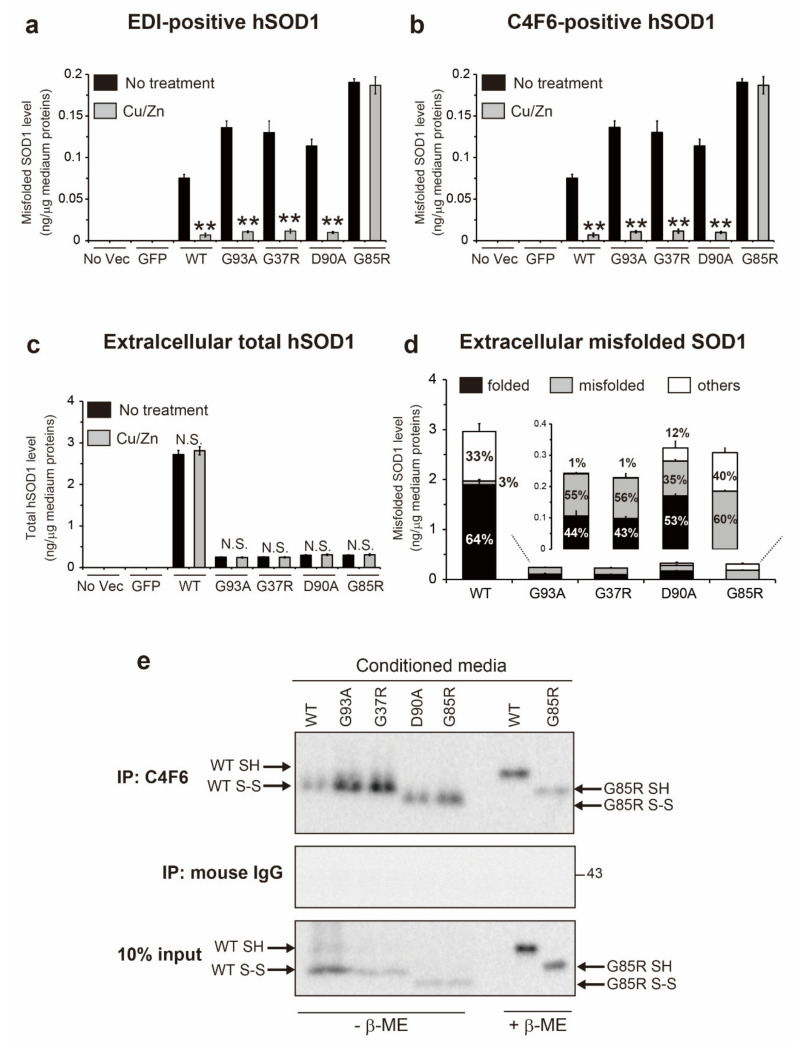
Extracellular misfolded human superoxide dismutase 1 (hSOD1) presents as a metal-free, disulfide oxidized form (apo-SOD1^S-S^). Enzyme-linked immunosorbent assay signals of (**a**) EDI-positive misfolded SOD1; (**b**) C4F6-positive misfolded SOD1; (**c**) total hSOD1, in the conditioned medium treated with or without both 1 mM CuSO_4_ and 1 mM ZnSO_4_ for 24 h. Data are expressed as the mean ± SD, *n* = 3 for each treatment per each transfection. Statistical analysis was performed using a two-tailed unpaired Student’s *t* test. ** *p* < 0.01 vs. Cu/Zn-untreated medium that corresponds to the same transfection. N.S., not significant; (**d**) the extracellular levels of different folding states of hSOD1 in the mediums; (**e**) extracellular misfolded SOD1 in the medium was detected by immunoprecipitation with C4F6 followed by non-reducing Western blotting to determine the thiol/disulfide redox balance of the misfolded species. β-ME = beta-mercaptoethanol.

**Figure 6 ijms-22-04155-f006:**
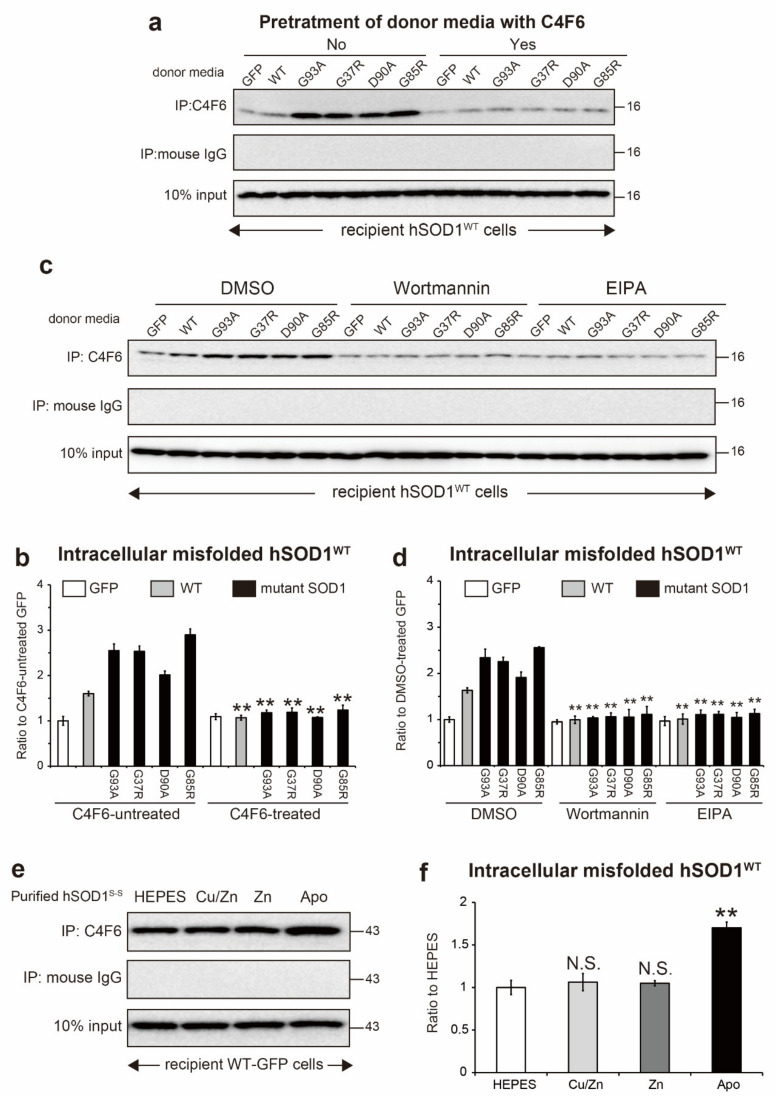
Extracellular misfolded apo-SOD1^S-S^ induces intracellular propagation of hSOD1^WT^ misfolding in recipient cells. A procedure to investigate intracellular prion-like propagation of human wild-type superoxide dismutase 1 (hSOD1^WT^) misfolding was performed using immunoprecipitation of the Nonidet P (NP-40) soluble fraction with C4F6 followed by Western blotting with antibodies to (**a**,**c**) SOD1 or (f) green fluorescence protein (GFP). The conditioned media from NSC-34 cells transfected with hSOD1-GFP containing misfolded apo-SOD1^S-S^ were used as a donor medium, whereas NSC-34 cells transfected with GFP-untagged hSOD1^WT^ were used as a recipient cell. (**a**) The recipient cells were exposed to conditioned medium pretreated with or without C4F6 to remove misfolded apo-SOD1^S-S^; (**b**) relative level of intracellular misfolded hSOD1^WT^ in the recipient cells exposed to medium pretreated with or without C4F6, ** *p* < 0.01 vs. C4F6-untreated cells that correspond to the same transfection, *n* = 3 for each treatment; (**c**) the recipient cells were pretreated with endocytosis inhibitors, 50 nM wortmannin, or 100 µM 5-(*N*-ethyl-*N*-isopropyl) amiloride (EIPA), for 1 h, and the cells were exposed to the medium for 24 h; (**d**) relative level of intracellular misfolded hSOD1^WT^ in endocytosis-impaired cells, ** *p* < 0.01 vs. DMSO-treaded cells that were exposed to the corresponding medium, *n* = 3 for each treatment; (**e**) the recipient cells expressing GFP-tagged hSOD1^WT^ were exposed to 0.5 µg/mL purified hSOD1^WT^ proteins with a disulfide bond (Cu/Zn SOD1^S-S^, Zn SOD1^S-S^, and apo-SOD1^S-S^) for 24 h, *n* = 3 for each treatment; (**f**) relative level of intracellular misfolded hSOD1^WT^ cells treated with the purified hSOD1^WT^ protein, ** *p* < 0.01 vs. HEPES-treated cells, N.S. = not significant. All data are expressed as the mean ± SD. Statistical analysis was performed using one-way ANOVA followed by the Tukey–Kramer post hoc test.

**Figure 7 ijms-22-04155-f007:**
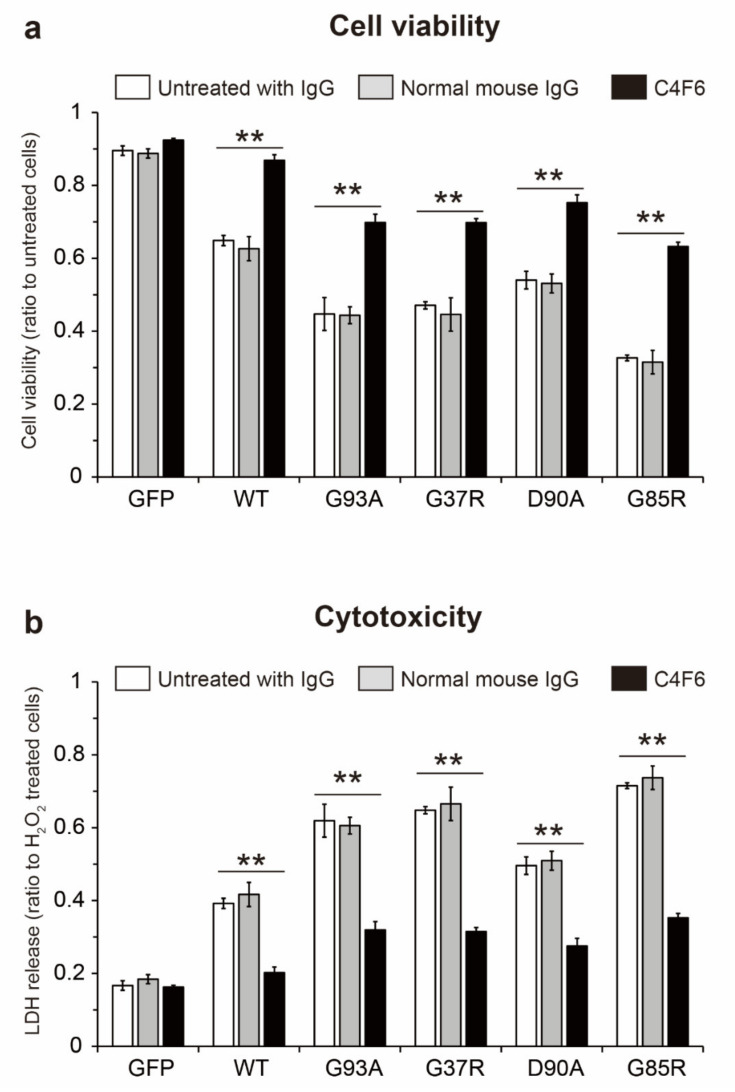
The conditioned medium containing extracellular misfolded metal free, disulfide oxidized superoxide dismutase 1 (apo-SOD1^S-S^) exerted cytotoxicity to motor neuron-like cells. (white) NSC-34 cells expressing green fluorescence protein (GFP) untagged human wild-type SOD1 (hSOD1^WT^) were exposed to conditioned medium from NSC-34 harboring hSOD1-GFP containing misfolded apo-SOD1^S-S^ for 24 h. Cell viability and cytotoxicity were assessed using (**a**) a Cell Counting Kit 8 assay and (**b**) a lactate dehydrogenase (LDH) release assay, respectively. (black) Extracellular misfolded apo-SOD1^S-S^ was removed from the conditioned medium by immunoprecipitation with C4F6. (gray) As a control, the medium was immunoprecipitated with normal mouse IgG. In the data set of cell viability, the results are expressed as the cell viability of recipient cells relative to that of untransfected cells, which were incubated with normal culture medium, Dulbecco’s modified Eagle’s medium and F-12 with GlutaMAX™ containing 1% (*v*/*v*) fetal bovine serum and 0.1 mM non-essential amino acids, instead of the conditioned medium. In the data set of cytotoxicity, the results are given as the amounts of released LDH to the medium relative to that of cells exposed to 100 μM H_2_O_2_ for 24 h. All data are expressed as the mean ± SD, *n* = 3 for each treatment. Statistical analysis was performed using one-way ANOVA followed by the Tukey–Kramer post hoc test. ** *p* < 0.01.

## Data Availability

The data presented in this study are available on request from the corresponding author.

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
