# Peer review of "A Metal-Free, Disulfide Oxidized Form of Superoxide Dismutase 1 as a Primary Misfolded Species with Prion-Like Properties in the Extracellular Environments Surrounding Motor Neuron-Like Cells"

_ijms, 2021, doi:10.3390/ijms22084155_

Round 1
Reviewer 1 Report
After 5 h of transfection with plasmid DNA, the Opti-MEM™ reduced serum medium was replaced with a mixture of 1:1 DMEM and Ham’s F-12 medium plus GlutaMAX™ (Gibco, Waltham, MA, USA) supplemented with 1% (v/v) fetal bovine serum, 0.1 mM non-essential amino acids, and 5 μM all trans-retinoic acid (Fujifilm Wako Pure Chemical Corporation, Osaka, Japan) for differentiation of the cells as described elsewhere [55,56].
None of the cited papers used the procedure as described in the manuscript. At least 4 days of RA treatment is required to achieve differentiation. The transfection with SOD1 should be performed using cells that were already differentiated. Otherwise, the authors cannot claim their work was done using differentiated cells. It makes no sense.
Response: We agree with your comment because the total amount of the ALS-causing SOD1 mutants in the conditioned medium is a 10-fold lower than that of wild-type SOD1. We understand that this trace amount of extracellular hSOD1 mutants is possibly difficult to see disulfide-reduced (SH) and -oxidized (S-S) form of the proteins. As you can see the below Figure A, we compared thiol/disulfide redox state of G93A SOD1, as a representative mutant SOD1 sample, using conditioned medium with or without concentration process. We did not observe the reduced form (SH) of G93A SOD1 even in the concentrated medium. Thus, we conclude that extracellular G93A SOD1 exists as a disulfide oxidized form (S-S). Based on the result, we used the original conditioned medium (unconcentrated medium) in this study to save our time of experiments for the concentration of the medium.
You should stress this point in the manuscript’s text.
Author Response
Minor revisions: ijms-1174137
Reviewer #1
We are most grateful to reviewer #1 for the critical comments and useful suggestions that have helped us to improve our manuscript. As indicated in responses that follow, we have taken all these comments and suggestions into accounts in the revised version of our manuscript.
Comment #1: After 5 h of transfection with plasmid DNA, the Opti-MEM™ reduced serum medium was replaced with a mixture of 1:1 DMEM and Ham’s F-12 medium plus GlutaMAX™ (Gibco, Waltham, MA, USA) supplemented with 1% (v/v) fetal bovine serum, 0.1 mM non-essential amino acids, and 5 μM all trans retinoic acid (Fujifilm Wako Pure Chemical Corporation, Osaka, Japan) for differentiation of the cells as described elsewhere [55,56].
None of the cited papers used the procedure as described in the manuscript. At least 4 days of RA treatment is required to achieve differentiation. The transfection with SOD1 should be performed using cells that were already differentiated. Otherwise, the authors cannot claim their work was done using differentiated cells. It makes no sense.
Response: We appreciate the detailed procedure for the differentiation of NSC-34 cells. According to your comment, our protocol for the differentiation of NSC-34 by treatment with all-trans retinoic acid may not be sufficient to mature all population of the cells and may result in the generation of heterogenous cell population: some cells are differentiated, but the remaining cells are still undifferentiated. We cannot provide evidence for that all of NSC-34 cells expressing the ALS-causing SOD1 mutants used in this study are certainly differentiated using our protocol. Thus, we should have removed the word of “differentiation” from the revised manuscript, and we have deleted two papers [#55: Maier et al. (2013) Neurochem Int 62 1029-1038; #56: Proctor et al. (2016) PNAS 113 614-619] from the References of the revised manuscript.
Comment #2: You should stress this point in the manuscript’s text. “We agree with your comment because the total amount of the ALS-causing SOD1 mutants in the conditioned medium is a 10-fold lower than that of wild-type SOD1. We understand that this trace amount of extracellular hSOD1 mutants is possibly difficult to see disulfide-reduced (SH) and -oxidized (S-S) form of the proteins. As you can see the below Figure A, we compared thiol/disulfide redox state of G93A SOD1, as a representative mutant SOD1 sample, using conditioned medium with or without concentration process. We did not observe the reduced form (SH) of G93A SOD1 even in the concentrated medium. Thus, we conclude that extracellular G93A SOD1 exists as a disulfide oxidized form (S-S). Based on the result, we used the original conditioned medium (unconcentrated medium) in this study to save our time of experiments for the concentration of the medium”.
Response: We agree that we should have explained why we used the original conditioned medium instead of the concentrated medium. In the revised manuscript, we have added the following sentence to the Results on page 8: “We confirmed that the disulfide oxidized form was a primary extracellular species of mutant SOD1 because the disulfide reduced form was not observed even in the concentrated conditioned medium”.
Reviewer 2 Report
In the very well written manuscript, the authors studied the properties of secreted WT and ALS-causing mutant human SOD1 protein in motor neuron-like NSC-34 cells. They identified that SOD1 mutation results in reduced secretion of the human protein (not murine SOD1) compared to WT, and that the mutant proteins displayed inferior activity/increased mis-folding in the medium. They then went on to show that apo-SOD1s-s, a metal-free and disulfide oxidized form of SOD1, is the main misfolded species in the condition medium of the NSC-34 cells. Apo-SOD1s-s in conditioned medium was subsequently convincingly shown to precipitate misfolding of SOD1 in a non-cell autonomous manner and to cause cytotoxicity and thus reduced cell viability.
The work is coherently and logically presented. The figures are clear and understandable. The conclusions are supported by the findings. It is hard to find fault with the manuscript. My comments are therefore very minor:
- The title is probably the weakest part of the manuscript – it is overly long and not particularly informative. A pithier title will attract more readers.
- A non-secreted control protein should be incorporated into the experiments presented in Figure 2, in order to assure the reader that the reported secretion is not simply caused by cell lysis (although unlikely).
- Please include reference in the Abstract to the cell line used in the study.
- In discussion of oxidative stress, the authors may wish to include reference to the work of Peter Oliver on Oxr1.
- The start of the sentence on Line 160 needs amending for clarity.
Author Response
We are grateful to see the attachment.

This manuscript is a resubmission of an earlier submission. The following is a list of the peer review reports and author responses from that submission.
Round 1
Reviewer 1 Report
This is a very well written manuscript, focusing on a very important topic. The authors show in a unique cell model (NSC-34) that G93A SOD1 mutation induce misfolded apo-SOD1S-S which may be link to the transmissibility of the disease.
The main problem of this article is the difficulty to generalized these results:I would have really appreciated that experiments where performed in parallel with two or more SOD1 mutations.
Moreover, I would have appreciated that previously published cell-model were also used.
Despite many qualities, I think that this article can only be accepted if additional experiments are made.
Reviewer 2 Report
The findings described in the manuscript are interesting, but there are several important concerns that are required attention:
1. Insufficient description of experimental procedures:
- It's unclear how the differentiation with retinoic acid was performed. From the text one can understand it was done by incubation with retinoic acid for 5h. No reference for this is provided. Most publications refer to 2 weeks of incubation with retinoic acid before the cells stop dividing (for example: DOI: 10.1016/j.neuro.2008.11.001).
- How the florescence microscopy of cells shown in FIG. 1 was performed.
474. "To extract proteins from NSC-34 cells, the cells were sonicated in ice-cold PBS (pH 7.0)". The conditions of the sonication are not described (and what instrument was used).
581. "incubated with a 1 M equivalent concentration of ZnSO4". Is it probably 1 mM?
2. Fig. 3A,B. The band of G93A is almost invisible. Thus, the determination of the ox/red ratio is not feasible. Why the authors didn't perform this experiment using the concentrated conditioned medium, whose preparation they described in the Methods.
3. It is not clear how the activity of murine SOD1 was determined in Fig. 4B. How was it extracted from the total activity corresponding to both human and murine SOD1 in the mixture?
4. In Fig 4C,D treatment with metal was performed in PBS buffer (10 mM phosphate), but this buffer in incompatible with divalent ions because insoluble salts are formed.
5. In Fig. 5F, what is the nature of the extra bands appearing in the WB of the conditioned media?
6. Fig. 6C. Why the intensity of the bands for GFP decreases in a way similar to both SOD1 proteins after the treatment with endocytosis inhibitors?
This particular experiment is very unconvincing, because the donor and recipient cell employed the same SOD1 construct: SOD1-GFP. In such settings, the risk of contamination of the recipient cells by the protein from the donor cell is very high. This experiment should use different constructs for the donor and recipient cells, to enable their separation on the gel, let's say SOD1 and SOD1-GFP or SOD1-GFP and SOD1. Thus, prion-like effect could be distinguished from a contamination.
7. Fig. 6B,D. The proteins are labeled wrongly in the legend.
258. "SOD1 species were not present, at least in part, in the conditioned medium from our motor neuron model of ALS".
What is the meaning of " at least in part" in this sentence?